# Design and Realization of Rural Environment Art Construction of Cultural Image and Visual Communication

**DOI:** 10.3390/ijerph20054001

**Published:** 2023-02-23

**Authors:** Fulong Liu, Baogang Lin, Kun Meng

**Affiliations:** 1College of Art, Xi’an University of Architecture and Technology, Xi’an 710055, China; 2College of Art and Design, Shaanxi University of Science and Technology, Xi’an 710021, China

**Keywords:** cultural image, rural environment, art construction, applied research

## Abstract

The practice of rural construction has been exploring and trying to adapt to the needs of rural development in various periods. In recent years, under the attention and promotion of the central policy, various social forces have joined the ranks of rural construction, and art intervention in rural construction has begun as a new method. Entering the public eye, it deeply intervenes in the construction and development of the countryside in a more gentle way, from the key point of interaction between the social and cultural orientation and the material needs of the countryside. However, most of the art interventions in rural construction practice only unilaterally use artistic techniques to beautify local areas or display works, without realizing the hidden artistic and cultural value of the village and ignoring the participation and role of the villagers in the whole process. After the construction is completed, once the foreign construction forces are withdrawn, the development of the village will stagnate. Therefore, mobilizing the main body of rural construction (original villagers) to participate in the joint construction of the village is an important link to solve the current problems of art intervening in the construction of rural settlements.

## 1. Introduction

At the beginning of the new journey of building a well-off society in my country, the rural foundation was weak. At the Fifth Plenary Session of the 16th Central Committee of the Communist Party of China, the goal of building a new countryside was put forward—production development, affluent living, civilized rural customs, clean and tidy village appearance, and democratic management. Entering a new era of socialism, in 2020, when the first centenary goal is about to be achieved, the main social contradictions have changed. Compared with the general requirements for the construction of new countryside at the Fifth Plenary Session of the 16th Central Committee of the Communist Party of China, in addition to “township and civilization”, the changes in the deepening of connotation are reflected for the first time. Rural development has risen to a strategic height. In recent years, research on rural revitalization has shown a spurt of growth [1]. Traditional villages have paid a large price in the transformation and development of modernization. The 19th National Congress of the Communist Party of China provided an important action guide for rural development in the new era, pointing out that rural development should implement the rural revitalization strategy. Sociologists, artists, local sages, and other experts from all walks of life have also invested in rural construction projects. Various projects related to art rural construction are also in full swing. With the advancement of new urbanization, the state pays great attention to rural issues, rural economy, and rural construction [2]. Cognition and thinking should be a special study that combines rural art practice with the specific concept of rural landscape sites [3].In the context of the rapid development of rural revitalization in China, the homogeneity of its cultural and environmental perception products is relatively serious. The article analyzes the cultural image and the professional method of visual communication in the rural environment, integrates the construction of the rural environment of the plastic arts intervention institution, and will ultimately bring you to experience the spiritual meaning of local culture in the environmental perception, leading you to have the aesthetic feeling of the scenery.

## 2. Relevant Theoretical Basis

### 2.1. Rural Revitalization

A village refers to a village or village settlement, a regional complex with natural, social, and economic characteristics, and has multiple functions such as production, life, ecology, and culture. The rural revitalization strategy is the strategy put forward by President Xi Jinping in the report of the 19th National Congress of the Communist Party of China. Efforts will be made to solve outstanding problems such as insufficient and comprehensive rural development. At the same time, the rural revitalization strategy is proposed on the basis of a profound understanding of China’s urban-rural relations, evolution trends, and laws. 

### 2.2. Art Creation

The modern Chinese dictionary explains the word “intervention” in this way: intervention means going deep into it and intervening [4]. The positive experience of real life endows the real significance of art intervention, which is mainly reflected in two aspects as shown in Table 1: on the one hand, art that can integrate its own life experience and care for real life is an art activity that can be inherited; “Art intervention” has the intervention of independent personality, the spirit of criticism, the courage to participate in society, promote communication, and enhance the sense of identity [5].

When art intervenes, the creator does not produce an “art product”, but forms a larger social event through the effectiveness of design art. They have an effective influence, causing people to think about related topics perception, so as to effectively intervene and reshape current social events. At the end of the 20th century, a community building movement emerged in Taiwan [6]. This community movement not only transforms the space environment, but also pays attention to the participation of community residents in public affairs, the happiness index in life, and the aesthetic perception of the community environment. The concept coincides with the concept of art intervention in rural construction under the concept of co-construction in this article. The community building of Tugou Village lets art connect residents’ lives and production spaces, so that villagers can feel the beauty brought by art to the community and actively participate in the public activities of building the community. The process of art entering into rural environment construction is shown in Figure 1:

### 2.3. Rural Landscape Environment

Rural landscape can be defined as a landscape space with human settlements and related behaviors in addition to urban landscapes. The core of rural landscape is rural settlement landscape. The landscape environment complex involves the individual architectural features, house structure, settlement pattern, etc., of rural settlements. It involves the environmental characteristics of the external space of the settlement, including the connection system with the external landscape environment [7]. In this study, the rural landscape environment with artistic intervention is a sustainable rural landscape environment that has an efficient artificial ecosystem, maintains the integrity and diversity of natural landscapes, and can inherit rural regional culture and a good rural living environment. In the face of the traditional architecture and landscape pattern of traditional villages being eroded by modern styles, traditional culture has been broken by modern life, and handmade products have lost their status as necessities of life and have been replaced by industrialized and machined products. Along with all these factors, the inherent living soil possessed by the countryside is losing its inheritance. It is in the embarrassing situation of being assimilated by urban culture.

### 2.4. Jointly Create Ideas

Based on the “people-oriented” living environment construction theory, the Ministry of Housing and Urban-Rural Development creatively put forward the concept of joint construction in response to community governance issues. The Ministry of Housing and Urban-Rural Development officially clarified the jointly created theory and implementation model in relevant documents and promoted the transformation of environmental construction from government-led to multi-party participation. In 2014, the concept of co-creation was applied to practice for the first time in the reconstruction and construction of old urban communities in Xiamen and achieved remarkable results.

Participation is the essence of co-creation. The innovation of this concept lies in the formation of a new governance model with villagers as the main body, decision-making, development, construction, construction, effect evaluation, and achievement sharing to solve the stubborn problems that hinder rural development. In this whole process, the status and role of the government and designers have changed compared with the past. They are no longer decision-makers, but the main participants, in guiding villagers to build beautiful homes. The application of the concept of co-creation in rural construction is conducive to accelerating the process of rural revitalization. The most important thing is to arouse the villagers’ subject consciousness, which is the key to solving the problem of rural development.

## 3. Research on the Application of Cultural Intention and Visual Communication Design in Rural Environment

### 3.1. Systematic “Five Blessings” Cultural Image

The application of cultural intention and visual communication design in rural environment is very extensive. Taking Zhanzishi Village in southeast Hubei as an example, the cultural intention of “Five Blessings” has been applied in various fields such as architecture and tourism. In the process of the construction of Chinese rural culture and local natural environment, the “auspicious” psychological state is widespread. Integrating the interpretation of the regional cultural markers of Zhanzishi Village, its regional culture has an inherent systematization. The local natural environment culture shows the comprehensive traditional Chinese “five blessings” culture. This value concept was handed down from Chongqing’s historical time and is not a reflection of the local living subject’s behavior norms. It forms a cultural artistic conception of “Shun”, “De”, “Gui”, “Fu”, “Shou”, and one by one. For example, the sweet persimmons match everything well and the number one bridge and the number one Lang pavilion match the spirit essence of the ten years of hard study. The capital bridge represents the pursuit of perfection of capital, the good official bridge metaphor the official life, the immortal bridge metaphor refers to the cultivation of immortality, which has extremely rich and very practical content.

### 3.2. Organization of Cultural System: Construct the Sequence of Virtual and Real Space

The comprehensive “Five Blessings” cultural artistic conception of Zhanzishi Village is integrated and the bridge section organization strategy of its garden landscape spatial sequence is carried out in the following manner: 1. Refine a clear cultural theme style and make it a clue to the structure of institutional environment art space; 2. The design scheme has an ingenious rhythmic sequence. This is similar to the traditional garden design principle of “layout planning”. 

The aesthetic sequence rhythm of natural environment space is regularly changed and reflected by the cultural elements or cultural modules in each space according to the designer’s orientation. The connection point of environmental art and the base price of the art site are set up according to the “life situation” and the traditional “five blessings” culture in the local culture. In the space language described by the regional culture, the Xianren River in Zhanzishi Village is the boundary of Wu Haiying. The south bank is the noisy vulgar, and the south bank is the spiritual spectacle. The four roads (“virtue”, “expensive”, “rich”, and “smooth”) all take the four “bridges” in the Xianren River as the starting point to open the different life realm of life. When we are accompanied by the winding rise of the countryside, the natural scenery along the line implies the warning of life, and there will be some fatigue and embarrassment along the way. The design of tourist attractions along these four roads focuses on the movement track of life. When reaching the end point, you can only choose one road (“longevity”) to go out of the mountain. It also implies that it is inevitable: however, it is not a pity, and life is not happy. 

## 4. Problems in the Creation of Rural Environmental Art

### 4.1. The Main Needs and Artistic Input Are Not Equal

Differences in growth environment, knowledge structure, artistic accomplishment, etc., lead to different attitudes of artists and villagers towards the countryside. Artists often consider how to activate the rural history and culture through their own creations, so as to reflect the cultural value of their works. It is easy to forget that the villagers are the creators of local culture, ignoring the subjective status of the villagers [8]. Therefore, it is difficult for artists to create art based on the actual needs of the villagers, which to a certain extent, discourages villagers from having positive attitudes towards art intervention. As the cultural level and aesthetic level are not generally high, the art creations performed by artists in the villages are only an “unknown way” to increase income in the eyes of the villagers. It is difficult for the villagers to participate in the project and create together with the artists. Art intervenes in the countryside to activate culture. The ideal is always the “wishful thinking” of the artist. In recent years, the Chinese government has vigorously advocated the protection and planning of traditional historical villages. More and more villages have been included in the list of protection and planning, as shown in Table 2:

### 4.2. Insufficient Participation of Villagers

The lack of attention to the subjectivity of the villagers makes the reality of rural development and artistic input unequal, which in turn, leads to a very low participation of villagers in the process of artistic intervention [9]. In the final analysis, art intervention in rural construction should take villagers as the main body and give full play to the wisdom and creativity of villagers. Participation methods also need to be improved. The current organizational form is shown in Table 3 below:

## 5. Research Results and Suggestions

### 5.1. Research Results and Reason Analysis

#### 5.1.1. “Source”—Extraction of Original Rural Site Information

The countryside is the birthplace of traditional culture. After the intervention of art in the construction, to a certain extent, cultural customs have been restored and continued, and the sustenance of rural culture and emotions has been preserved. It is also a way to show rural history and culture to the new generation of people. The rural environment has attracted urban residents to visit because of the intervention of art and the charm of the countryside. Tourists see the brilliance of traditional farming culture in the new era. Not only did the countryside win attention, but the rural culture was also valued. Villagers also felt different spaces and concepts due to the intervention of art [10]. Art is the best glue between urban and rural areas. It allows urban and rural cultures to communicate with each other, allowing art to penetrate deeply into the countryside and make it regain its brilliance.

Where does the “source” come from? It can be understood as the “raw material” of the countryside. When art intervenes in the countryside, artists have already appeared here, and the scene needs to exist. The scene is equivalent to the identity of “raw material” to a large extent. The artist needs to obtain information from the “source”, which is the artist’s creative method and inspiration source. In the process of the artist’s creation, there will be two “source” creation paths. One is that the artist abandons the extraction and refinement of the “source” and insists that his own creation has nothing to do with the original venue and audience; here, the artist does not care about the original source. The venue is dynamic. On the other hand, the artist’s creative path reflects that some artists insist that, in their artistic creation, they must understand the local situation, have a thorough understanding of the “source”, and extract the corresponding cultural element symbols from it. Then, the direction of artistic creation can be determined: see Figure 2.

#### 5.1.2. “Sink”—The Output of Rural Site Information

In the process of rural construction, traditional rural residential buildings are often considered to be “shabby houses” and are demolished. The effect of the construction no longer has the taste of the countryside and loses the traditional cultural characteristics of the countryside. Under the concept of co-construction, art intervention in the rural environment is not a cultural invasion, but a benign guidance [11]. Artists and designers should gradually change the villagers’ perception of rural things through the intervention of art, have a certain understanding of the rural aesthetic standards, and learn how to express their own aesthetic ideas. When the overall aesthetic level of the village is improved, the traditional culture of the village can be continued well.

The output of “Hui” about rural site information also means that we need to think about what kind of products we can provide when designing and creating. Taking the land landscape as an example, the vast rural land is increasingly favored by artists to create land art. This is a transformation of the content of the rural space after the intervention of art. The vast land has been freed from agricultural production. The environment is bound, and the remodeling into a work of art through design art should be promoted. In the work of art, the environment itself must be purified. The landscape should also be purified to slightly weaken the things that are too artificial. 

#### 5.1.3. “Field”—To Create the Presence of Rural Space

The place represents the viewpoint and vision of the rural users. The place shows memory through time and changes, and time will leave a memory in the spiritual space of the place [12,13,14,15]. When a work of art is involved in rural practice, the external material and internal spiritual connotation depend, to a large extent, on the particularity of the realized spatial landscape environment. The spatial scale, spatial texture, and local materials in the rural landscape environment all play an important role. The symbolic value in the rural place reflects the accumulation of spiritual connotation value and represents the rural spiritual place. The spirit of the place in the spiritual place carries the sense of belonging and identity in the rural space and place. The type of field is shown in Figure 3.

Farming is the most traditional way of production in rural areas, and agriculture has been a pillar industry in Chinese rural areas since ancient times. However, after the social development and progress of today, farmers are no longer willing to go to the fields; relying on growing crops cannot bring better economic benefits. The traditional rural farming culture is in crisis. Should art intervene in the countryside, and should it change the rural status quo of the industrial structure dominated by agriculture? The answer is negative. When art intervenes in rural construction, we should first protect the traditional rural farming culture. Then, under the guidance of the concept of joint construction, we should combine the secondary and tertiary industries to realize the linkage of multiple industries, coordinated development, and improve the endogenous development momentum of rural areas. This should be done to create their own industries and brands and integrate art into the rural environment: see Figure 4.

### 5.2. The Design Method of Art Intervening in Rural Environment

#### 5.2.1. Co-Participation in the Design Process—Bottom-Up, Multi-Collaboration

In the past, when art intervened in rural environment construction, artists or designers led the construction plan. However, farmers hardly participated in the entire construction process [16]. Although this “all-in-one” intervention method can quickly improve the appearance of the village, it cannot stimulate the villagers’ sense of belonging and responsibility to their homes. In the later maintenance stage after the construction is completed, it is difficult for the villagers to spontaneously maintain the construction results and buildings. The effect is far less effective than expected. Under the concept of co-construction, art intervention in rural environmental construction must change the conventional construction method, guide villagers to actively participate in the construction of their homes, allow designers to participate in design guidance and services throughout the process, and form a bottom-up design process model with the government. More attention should be paid to the cultivation of the participation process, so that the villagers can truly participate in the construction of their homes.

The government, designers, and villagers participate in the whole process of the construction process. First, they can understand the advantages and disadvantages of the village and the actual needs of the villagers in more detail. Then, they can have an impact on the direction of the village’s development. The second is to better coordinate the relationship between the government, designers, villagers, social forces, and other parties. Third, it can not only complete the transformation of environmental space, but also promotes industrial upgrading, enhances the self-governance ability of the village, and finally realizes sustainable development of the village.

#### 5.2.2. Common Progress of Design Concepts—Artistic Vision of the Countryside

The difficulty of rural construction lies in “people”, and the difficulty of art rural construction is more prominent in “people”, because there have always been different opinions in terms of aesthetic awareness. The aesthetic concepts of villagers and artists are often completely different, and their ideas are inconsistent. How can they move towards the same goal? The first step in the design of rural environment under the concept of joint construction is to change the concept of villagers, look at the countryside from an artistic perspective, re-understand the value of the countryside, and re-understand the wisdom of the countryside [17]. The success of Japanese art village construction lies in the fundamental change in the concept of farmers.

Rural revitalization is essentially rooted in the cultural customs, the way of life, and the villagers’ inheritance of traditional handicraft skills. The seemingly random and irregular construction in the countryside contains the traditional construction techniques of local dwellings and conveys the cultural attributes and regionality of the village itself. Craftsmen with traditional craftsmanship in the countryside use their own craftsmanship and wisdom to build village houses, manufacture living utensils, and connect rural life. Before the reconstruction of the village, it is necessary for artists, designers, and villagers to regain the wisdom of the functional layout of the rural space and the traditional rural craftsmanship to awaken the rural memory. In my country’s traditional villages, the spatial texture of the village looks natural, but it is also the most reasonable. The location of each building, the location of the village entrance, and the location of the ancestral temple have their own rules of development. From any angle, the wisdom of the countryside is revealed [18].

#### 5.2.3. The Beginning of the Change of Design Concept—Starting from the Needs of the Villagers

In the previous construction, designers and artists simply took “beautiful and livable” as the goal of village construction. They think that as long as the living space environment of the villagers is improved, the living standards of the villagers can be improved, and the village can develop. However, only focusing on the “appearance” of rural construction cannot make the road of rural construction go very far, and some villages have even stopped developing. To realize the transformation from “beautiful and livable” to sustainable development of rural construction, designers and artists need to start from the actual problems of the village, change their design thinking, and explore more effective construction methods for the countryside. Starting from the foundation of rural industries, creating opportunities for industrial development created by multiple functions is a necessary condition to stimulate the enthusiasm of village development. According to the actual conditions of different villages, the feasibility of rural industry cultivation should be investigated, existing industries should be continued, and new industries that can be cultivated should be added; this will stimulate the endogenous power of village revitalization. 

Rural environment designers, under the guidance of the concept of co-construction, should abandon the ideal thinking of rural construction, listen to the real voice of the village, face the actual demands of the villagers, and let the designers truly serve the village. In this way, the enthusiasm of the villagers will be mobilized and devoted to home construction, as shown in Figure 5.

#### 5.2.4. Mutual Transfer of Design Ideas—An Easy-to-Communicate Way of Expression

In the past, in cases of art intervention in rural environment construction, participation of villagers in the design stage was very low. One of the important reasons was that the ideas of designers, artists, and villagers were not well communicated to each other. Designers and artists cannot stand in the position of villagers, and the villagers cannot understand and accept the ideas of designers and artists. Therefore, it is important for art to intervene in the rural environment under the concept of joint construction to make it easier for both parties to obtain each other’s ideas and information [19].

The common way of expressing the environmental design scheme is to use a computer to construct a site model and to render the design. This method is suitable for use in the city, but it does not seem to be the best method for the countryside. The more acceptable way of expressing drawings for villagers is the complete integration of the design scheme and the actual site. The designer draws the design ideas directly from the photos of the original site and compares them before and after the renovation. This reflects the design effect more intuitively [20].

## 6. Conclusions

Taking Zhanzishi Village in southeastern Hubei Province as an example, this paper analyzes the purpose of cultural intention and visual communication design of the rural environment. On the one hand, the village environment is beautiful, and the villagers’ happiness index is improved; On the other hand, we should develop rural tourism and improve the economic situation of the villagers. In the new era, the purpose of rural environmental art design is to improve the quality of rural residential environment. With the strong development trend of Chinese cities, rural areas are slowly trapped in green ecological dilemma and aesthetic misconceptions. In many places, the problem of simplification is more serious in the process of development, ignoring the establishment of the existing rural aesthetic situation and the overall level of rural environmental art design in the new countryside. As the level of global integration and economic development globalization continues to increase, the traditional farming culture and technology in China have been greatly affected. The inadequate and uneven development of regions and cities has seriously affected the rural living environment, and traditional rural aesthetic culture and art have been severely tested. “Beautiful rural construction” should be realized from time to time to inject new vitality into the new rural infrastructure, generally learn the experience and lessons of urban infrastructure, and use visual communication art introduction to highly combine aesthetics. It can improve the quality of rural plastic arts, the natural environment, and technology on the premise of meeting the growing material civilization and cultural needs of farmers. Therefore, the new rural environmental art design and visual communication must actively refer to the experience and lessons of the city. It should use the improvement of local comprehensiveness as a guidance, reflect the progressive atmosphere of the times, and promote the effective development of the new rural environmental art design and visual communication work. This should be done on the premise of meeting the chemical life vision of farmers.

## Figures and Tables

**Figure 1 ijerph-20-04001-f001:**
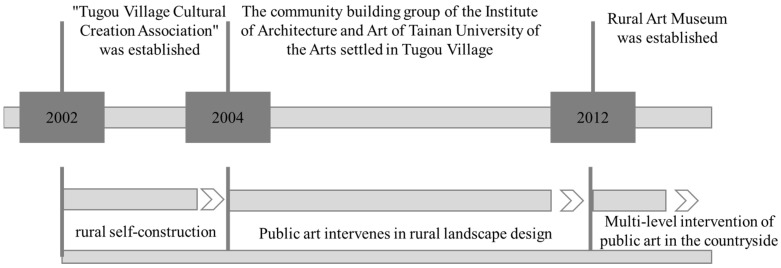
The process of rural environment construction.

**Figure 2 ijerph-20-04001-f002:**
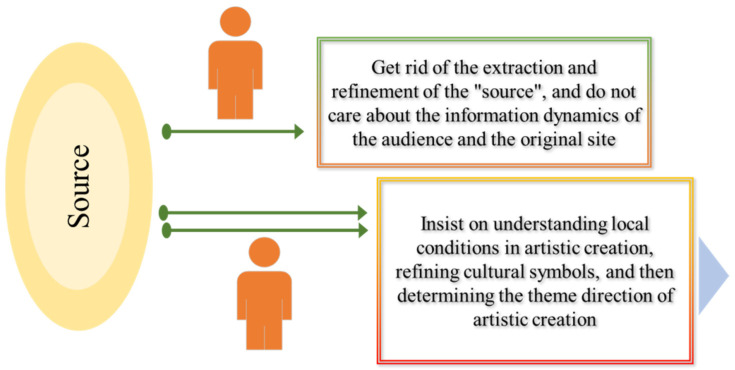
“Source” authoring path.

**Figure 3 ijerph-20-04001-f003:**
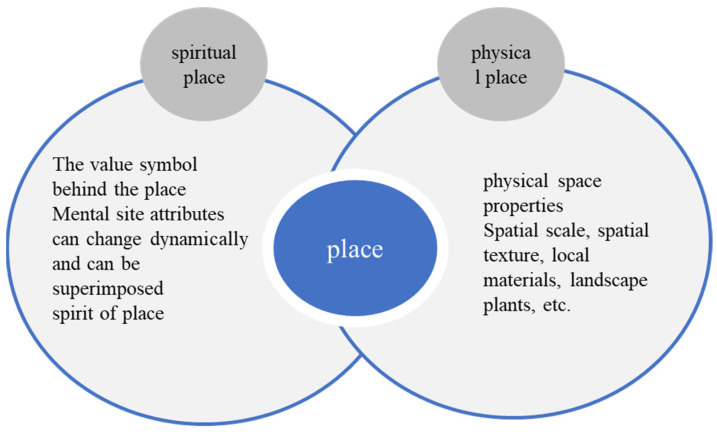
Types of “fields”.

**Figure 4 ijerph-20-04001-f004:**
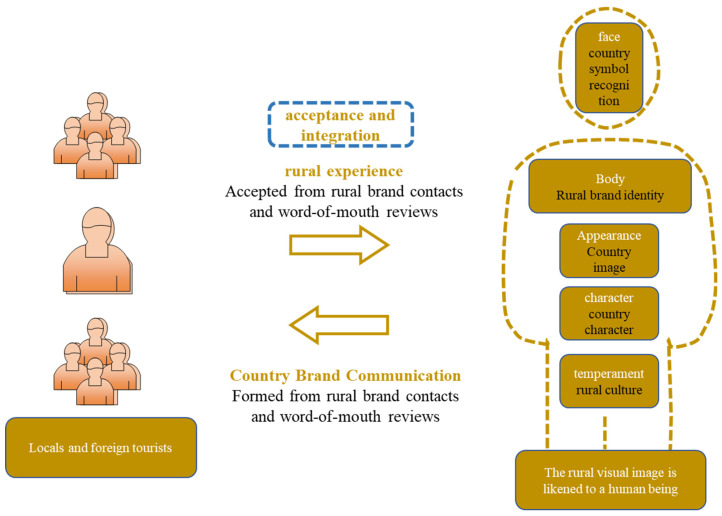
Country brands.

**Figure 5 ijerph-20-04001-f005:**
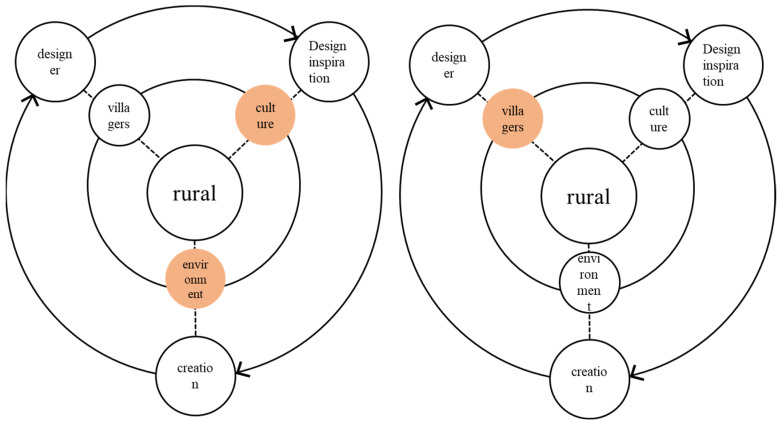
Changes in designers’ rural construction ideas.

**Table 1 ijerph-20-04001-t001:** Two aspects of artistic involvement.

Dimensions	Intervention Features
Dimensions in terms of art	Gives art a greater possibility of inheritance
Life aspects	Make life more artistic

**Table 2 ijerph-20-04001-t002:** Preparatory stage.

Preliminary Preparation Steps	Project Approval	Event Planning	In-Depth Research	Information Collection	Data Analysis
Participating subject	Government, designers, villagers	Government, designers, villagers	Government, designers, villagers	Government, designers, villagers	Analyst
Main content	In view of the goal of village construction, all parties discussed together	Before entering the village to carry out work, carry out a series of publicity activities that can bring the relationship between the villagers closer together	Comprehensive and in-depth understanding of various elements in the countryside, including environment, architecture, villagers, production, and living organization	Organize and record research data	Objective and comprehensive analysis of survey data
Work method	Hold a meeting	Public welfare activities, volunteers in the village to explain	Interviews, questionnaires, surveys, participatory observation, developing detailed survey checklists	Drone shooting, GIS geographic information system, on-site photo shooting	SWOT analysis, chart analysis

**Table 3 ijerph-20-04001-t003:** Comparison of the process of rural construction with traditional art and rural construction under the concept of co-construction.

Compare Content	Traditional Art Intervenes in Rural Environment Construction	Artistic Intervention in Rural Construction under the Concept of Joint Construction
Participating subject	Government, artist, designer	Villagers, government, artists, social institutions, village organizations
Way of participation	Top down	Bottom-up, diverse collaboration
Participation time	Not stationed in the village	Resident in the village, participating in the whole process of construction

## Data Availability

The labeled data set used to support the findings of this study is available from the corresponding author upon request.

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
