# Peer review of "Design and Realization of Rural Environment Art Construction of Cultural Image and Visual Communication"

_ijerph, 2023, doi:10.3390/ijerph20054001_

Round 1
Reviewer 1 Report
The present manuscript argues that art interventions in rural environments should be participatory. If participation is not achieved, then it is purported that stagnation will occur. The hypothesis is valid, as phenomena of this nature can be observed in many countries.
The introduction states China as the place of observation. Then, on line 66, it is situated more specifically in Taiwan. Are we on the results section at this point? The only reference in this paragraph is from research done in Germany.
Lines 56-57: Intervention should not be defined tautologically, that is, by using the word “intervene” in the definition.
After various definitions and theoretical arguments, the results seem to begin on page 113. No objective for the research is stated, nor is there an explicit methodology made clear.
The main results are basically two:
1. Artists do not consider the needs of the villagers
2. There is insufficient participation of villagers in terms of artistic input.
These two results are basically the same. The only evidence for the results are two tables, purportedly from the Government but with no source included.
On line 142, a section called “Reasons and recommendations” begins. Are these results? Line 142, states that these are supposed to be results, but more theory is what follows. The citation on line 152 is from Spain, not from China.
Conclusions begin on line 298. It presents evidence from a harvest festival in China that was not explained in the results section. And that´s how it ends, with an explanation of what happened at the festival.
Line 320 states that there is labeled data to support the findings.
In sum: The manuscript does not merit publication for the following reasons: There is no clear objective, the manuscript is not placed in a line of research, there is no clear method for data analysis, there is virtually no data presented, and there are no general conclusions that are supported by the data.
Author Response
Dear Reviewer, I have revised the mansucript according to your comments, please see the attached file for every reply to your comments.

Reviewer 2 Report
The authors address an interesting topic.
This is a disclosure document, not a research article. If what you aspire to is the latter, I recommend that authors analyze the characteristics and structure of this type of document beforehand.
At the European level, there is an extensive scientific literature that addresses the impact of developmentalism on heritage and rural societies. A detailed review of this matter with an international vision beyond the Chinese experience would be enriching.
The research objectives are not defined; that is why the conclusions cannot be related to them. They may not have it with the document as a whole either.
Author Response
Dear Reviewer, thanks for your comments, I have made revisions accordingly, please see the attached file.

Reviewer 3 Report
It is advisable to better describe the methodology used, extend and argue the conclusions
Author Response
Dear Reviewer, thanks a lot for your comment, here's my reply: This paper has been written by studying several examples, and the conclusions have been rewritten. With best regards.
Round 2
Reviewer 1 Report
Although the main argument is valid, there is no clear methodology or sufficient impartial evidence to support it. The definition of intervention is still tautological. There is no clear research objective. The conclusion is better than in the last version, but there is no substantial scientific evidence to support it.
Author Response
As for the definition and description of artistic intervention, I have checked various translation methods. If the reviewer has a better explanation, please correct it. The research objectives have been marked in blue in the introduction. The article lists the example of Chian Zishi village in southeast Hubei (green font) to further verify the conclusion and mark it with yellow font.
Reviewer 2 Report
I confirm the indications given in the first round of review. Apart from merely formal touch-ups, the article needs deep and structural changes.
Author Response
The structure of the article has been readjusted: introduction - explanation of theoretical concepts - current situation of the application of cultural intention and visual communication design in rural environment - existing problems at this stage - measures to solve the research results - conclusion.